# Trifluoperazine, an Antipsychotic Drug, Effectively Reduces Drug Resistance in Cisplatin-Resistant Urothelial Carcinoma Cells via Suppressing Bcl-xL: An In Vitro and In Vivo Study

**DOI:** 10.3390/ijms20133218

**Published:** 2019-06-30

**Authors:** Kuan-Lin Kuo, Shing-Hwa Liu, Wei-Chou Lin, Fu-Shun Hsu, Po-Ming Chow, Yu-Wei Chang, Shao-Ping Yang, Chung-Sheng Shi, Chen-Hsun Hsu, Shih-Ming Liao, Hong-Chiang Chang, Kuo-How Huang

**Affiliations:** 1Graduate Institute of Toxicology, College of Medicine, National Taiwan University, Taipei 100, Taiwan; 2Department of Urology, College of Medicine, National Taiwan University, Taipei 100, Taiwan; 3Department of Urology, National Taiwan University Hospital, Taipei 100, Taiwan; 4Department of Medical Research, China Medical University Hospital, China Medical University, Taichung 404, Taiwan; 5Department of Pediatrics, National Taiwan University Hospital, Taipei 100, Taiwan; 6Department of Pathology, National Taiwan University Hospital, Taipei 100, Taiwan; 7Graduate Institute of Clinical Medicine, College of Medicine, National Taiwan University, Taipei 100, Taiwan; 8Department of Urology, New Taipei City Hospital, New Taipei City 112, Taiwan; 9Graduate Institute of Clinical Medical Sciences, College of Medicine, Chang Gung University, Taoyuan 333, Taiwan

**Keywords:** urothelial carcinoma, trifluoperazine, chemotherapy resistance, Bcl-xL, apoptosis

## Abstract

Cisplatin-based chemotherapy is the primary treatment for metastatic bladder urothelial carcinoma (UC). Most patients inevitably encounter drug resistance and resultant disease relapse. Reduced apoptosis plays a critical role in chemoresistance. Trifluoperazine (TFP), an antipsychotic agent, has demonstrated antitumor effects on various cancers. This study investigated the efficacy of TFP in inhibiting cisplatin-resistant bladder UC and explored the underlying mechanism. Our results revealed that cisplatin-resistant UC cells (T24/R) upregulated the antiapoptotic factor, B-cell lymphoma-extra large (Bcl-xL). Knockdown of Bcl-xL by siRNA resensitized cisplatin-resistant cells to the cisplatin cytotoxic effect. TFP (10–45 μM) alone elicited dose-dependent cytotoxicity, apoptosis, and G0/G1 arrest on T24/R cells. Co-treatment of TFP potentiated cisplatin-induced cytotoxicity in T24/R cells. The phenomenon that TFP alleviated cisplatin resistance to T24/R was accompanied with concurrent suppression of Bcl-xL. In vivo models confirmed that TFP alone effectively suppressed the T24/R xenograft in nude mice. TFP co-treatment enhanced the antitumor effect of cisplatin on the T24/R xenograft. Our results demonstrated that TFP effectively inhibited cisplatin-resistant UCs and circumvented cisplatin resistance with concurrent Bcl-xL downregulation. These findings provide a promising insight to develop a therapeutic strategy for chemoresistant UCs.

## 1. Introduction

Bladder urothelial carcinoma (UC) is the sixth most common cancer in the United States, with approximately 74,000 predicted new cases in 2015 [1]. UC constitutes more than 90% of bladder cancers. Despite radical cystectomy, approximately 50% of cases of high-grade and muscle-invasive bladder UC progress to metastatic diseases. The standard therapy for metastatic bladder UC is cisplatin-based chemotherapy [2]. Despite initially exhibiting positive responses to chemotherapy, most patients experienced relapse and resultant mortality due to chemotherapy resistance. The prognosis of patients with metastatic UC has been ominous [3]. Developing strategies to circumvent chemoresistance to improve the outcomes of metastatic bladder cancer is imperative.

DNA-damaging agents, such as platinum-based chemotherapeutic drugs, are widely used in chemotherapy regimens. Cisplatin (cis-diamminedichloroplatinum II, CDDP) is the main component of chemotherapy for metastatic UCs. The antitumor mechanism of cisplatin involves the crosslinking with purine DNA bases, subsequently resulting in DNA adducts, which inhibit DNA replication and transcription [4]. Cancer cells respond to cisplatin-induced DNA damage by activating a network of damage response pathways that regulate cell cycle arrest, DNA repair, and apoptosis [4]. Multiple genetic and epigenetic factors can contribute to resistance of chemotherapy [5,6]. The suppression of apoptosis represents a key determinant of chemotherapy resistance, which has been attributed to altered expression patterns of antiapoptotic and proapoptotic proteins [7,8,9]. Among them, downregulated B-cell lymphoma-2 (Bcl-2) family proapoptotic proteins or upregulated antiapoptotic molecules, such as B-cell lymphoma-extra large (Bcl-xL), have been widely investigated [6,10,11,12].

Trifluoperazine (TFP), a phenothiazine derivative, has been commonly used as an antipsychotic drug. Previous studies demonstrated that TFP alone or combined with chemotherapy effectively induced tumor inhibition [13,14,15,16,17]; moreover, it could circumvent drug resistance in various cancers [18,19,20,21,22,23]. The mechanisms underlying the antitumor effect of TFP have been reported to be associated with anti-cancer stem cell properties by suppression of stemness-associated expression, such as CD133, c-Myc, β-catenin, and modulating apoptotic factors, including Bax, Bad, Bcl-2, and caspases or cell cycle arrest. However, the potential antitumor effects of TFP on bladder UCs remain unclear.

To bridge the research gap, we conducted an in vitro and in vivo study to investigate the efficacy of TFP to inhibit human cisplatin-sensitive and cisplatin-resistant UC cells. Moreover, we sought to explore the underlying mechanism involved in the TFP anti-tumor effect on cisplatin-resistant UCs.

## 2. Results

### 2.1. Cisplatin-Induced Cytotoxicity, Apoptosis, and DNA Damage Response Were Reduced in Cisplatin-Resistant UC Cells (T24/R) Compared with Parental T24 Cells

We first investigated the effects of cisplatin (10–40 μM) on viability and apoptosis of parental T24 and cisplatin-resistant subline (T24/R) cells. As illustrated in Figure 1A,B, cisplatin effectively induced cytotoxicity and apoptosis in T24 cells at 24 h after treatment. However, the cisplatin-induced cytotoxicity and apoptosis was significantly reduced in the resistant cells (T24/R) compared with the parental T24 cells. Consistently, T24/R cells demonstrated reduced levels of phospho-histone H2A.X, a DNA damage marker, compared to T24 cells after cisplatin treatment (Figure 1C).

### 2.2. TFP Effectively Induced Cytotoxicity, Apoptosis, Endoplasmic Reticulum Stress-Related Apoptosis, and DNA Damage in Cisplatin-Resistant Human UC Cells (T24/R)

We then examined cytotoxic and apoptotic effects of TFP on cisplatin-resistant UC cells (T24/R). As presented in Figure 2A, TFP effectively inhibited cell viability in a dose-dependent manner at 24 and 48 h. In addition, treatment with TFP (25 μM) for 24 h significantly induced apoptosis in cisplatin-resistant T24/R cells. The expression of cell stress markers (Phospho-SAPK/JNK), endoplasmic reticulum (ER) stress-related apoptosis proteins (CHOP and caspase-4), and a DNA damage marker (phospho-histone H2A.X) increased. Meanwhile, the anti-apoptotic molecule Bcl-xL decreased after TFP treatment.

### 2.3. TFP Induced G0/G1 Arrest in Cisplatin-Resistant UC Cells (T24/R)

A previous study reported that TFP caused cell cycle arrest at the G0/G1 phase [13]. We, thus, analyzed the effect of TFP on cell cycle progression of T24/R cells. Flow cytometry analysis revealed that 25 μM TFP-treated T24/R cells were blocked at the G0/G1 phase after 24 h (Figure 3A). Moreover, the expression levels of the cyclin-dependent kinase inhibitors, p21 and p27, increased at 24 h after TFP treatment (Figure 3B).

### 2.4. TFP Enhanced the Cisplatin Antitumor Effects and Alleviated Cisplatin Resistance with Concurrent Bcl-xL Suppression in T24/R Cells

Next, we evaluated the apoptotic and cytotoxic effects of TFP alone and in combination with cisplatin on T24/R cells using MTT assay and flow cytometry with propidium iodide (PI) and Annexin V-FITC staining, respectively. Due to chemo-resistance, cisplatin (10–50 µM) alone could not induce apoptosis and cytotoxicity in T24/R cells after 24 h of exposure (Figure 4A,B). Moreover, CalcuSyn software was used to analyze the combinative drug effect of TFP and cisplatin on T24/R cells. The combinative effects of TFP and cisplatin at the concentration ratio of 1:1.25 (TFP:cisplatin) were applied to the median-effect analysis using the mutually nonexclusive model. The combinative effect was then transformed into and presented in a median-effect plot, dose-effect plot, and fraction affected-combination index plot, as is shown in Figure 4C. The combination index of two drugs at the concentration ratio of 1: 1.25 was less than 1, which indicated a synergistic effect. However, TFP alleviated drug resistance of T24/R to cisplatin and enhanced the apoptotic and cytotoxic effects of cisplatin on T24/R cells (Figure 4A–C). Bcl-xL, an anti-apoptotic molecule, has been reported to govern drug sensitivity to chemotherapy in various malignancies [24,25,26,27,28]. Thus, we evaluated the differences in expression of Bcl-xL between parental cisplatin-sensitive T24 and cisplatin-resistant T24/R cells after cisplatin treatment. The results indicated that cisplatin increased the expression of Bcl-xL in both cell lines. The expression of Bcl-xL was more abundant in T24/R cells when compared to that in the parental T24 cells (Figure 4D). We then analyzed Bcl-xL expression after TFP treatment to elucidate the underlying mechanism of TFP to resenstitize 24/R cells to cisplatin treatment. In Figure 4E, co-treatment with 12.5 μM TFP suppressed Bcl-xL levels and activated phospho-histone H2A.X. We hypothesized that upregulated Bcl-xL was associated with cisplatin resistance in T24/R cells. TFP alleviated cisplatin resistance in T24/R cells and resensitized T24/R cells to cisplatin and was accompanied with the suppression of Bcl-xL. Furthermore, we used Bcl-xL siRNA knockdown to clarify the role of Bcl-xL in cisplatin resistance of T24/R cells. After treating T24/R cells with 10 nM Bcl-xL siRNA or non-targeting scramble siRNA as a control, we observed Bcl-xL knockdown decreased Bcl-xL levels and restored the cisplatin-induced DNA damage (phospho-histone H2A.X activation) and cytotoxicity (Figure 4F).

### 2.5. TFP Enhanced Antitumor Effect of Cisplatin in a Xenograft Mouse Model of Cisplatin-Resistant UC Cells (T24/R)

We then sought to confirm the antitumor effects of TFP in vivo by using a xenograft mouse model. T24/R cells were mixed with Matrigel and injected subcutaneously into the flanks of homozygous null (nu/nu) mice. The mice were divided into four groups. Mice were injected intraperitoneally with mock (nontreated control, *n* = 4), cisplatin (*n* = 4), TFP (*n* = 5), or cisplatin combined with TFP (*n* = 5) for 4 weeks, as described in the Materials and Methods section. TFP alone effectively suppressed T24/R xenograft in nude mice. Combined treatment with cisplatin and TFP exerted the most significant antitumor effect on T24/R xenografts when compared with those treated with cisplatin or TFP alone (Figure 5A,B). In addition, we observed that body weights and side effects among mice from the four treatment protocols did not show differences, which indicated that treatment with TFP and cisplatin did not produce any apparent toxicity in mice (data not shown).

## 3. Discussion

Cisplatin has been the primary constituent of standard chemotherapy regimens for treatment of metastatic UCs. However, its toxicity and the emergence of drug resistance have compromised its therapeutic efficacy. The mechanism of cisplatin-induced cytotoxicity is associated with the ability to crosslink DNA and induce DNA damage. DNA damage will activate the DNA repair system and cells undergo apoptosis if DNA repair fails [29]. Bcl-xL, a member of the Bcl-2 family, inhibits apoptosis responding to stress insult through two different mechanisms: by heterodimerization with an apoptotic protein to inhibit its apoptotic effect and by maintaining normal function of mitochondrial membrane to prevent the release of the caspase inducer by binding to the voltage-dependent anion channel. Downregulation of Bcl-xL was reported to reverse cisplatin resistance of cancer cells [7,8,9,30,31].

This is the first study to investigate the therapeutic efficacy of TFP on UCs and cisplatin-resistant UCs. We demonstrated that TFP effectively alleviated cisplatin resistance and enhanced cisplatin-induced cytotoxicity in cisplatin-resistant UCs in vitro and in vivo. Moreover, we observed cisplatin-resistant UC cells (T24/R) exhibited higher expression of Bcl-xL. Knockdown Bcl-xL by siRNA restored cisplatin-induced cytotoxicity and DNA damage in T24/R cells. A previous study indicated that TFP could inhibit the repair of bleomycin-induced DNA double-strand break and re-sensitize non-small cell lung cancer cells to chemotherapy [32]. Another study also showed that TFP upregulated the Bax/Bcl-2 ratio to promote apoptosis [21]. Consistent with these findings, we found TFP enhanced cisplatin-induced cytotoxicity and alleviated cisplatin resistance in cisplatin-resistant T24/R cells via suppression of Bcl-xL [18,19,20,21,22,23] with activation of DNA damage marker, phospho-histone H2A.X, and with concurrent downregulation of Bcl-xL.

Metastatic UC is a lethal disease, and platinum-based chemotherapy remains the standard of care in first-line therapy. The substantial chemotherapy-related toxicity and subsequent drug resistance led to treatment failure and ominous prognosis. For decades, other second-line therapies did not show the efficacy to improve the survival in such patients although immune checkpoint blockade is recently poised to change the treatment paradigm. This is the first study to demonstrate therapeutic efficacy of TFP on UCs and cisplatin-resistant UCs, which provided preliminary evidence for TFP as a second-line therapy after cisplatin-resistance. In addition, TFP and cisplatin are clinically approved drugs, and thus the combination therapy described herein could be fast-tracked into clinical trials. The results present important insight for further clinical applications.

## 4. Materials and Methods

### 4.1. Cell Culture

The T24 cell line, derived from a patient with grade III bladder urothelial carcinoma, was obtained from the Bioresource Collection and Research Center (Hsinchu, Taiwan). The cell line was cultured in RPMI-1640 medium supplemented with 10% heat-inactivated fetal bovine serum, 1 mM sodium pyruvate, and penicillin (100 units/mL)/streptomycin (100 μg/mL) at 37 °C with 5% CO_2_. All culture media and supplements were purchased from Invitrogen (Carlsbad, CA, USA). Cisplatin-resistant UC cells (T24/R) were derived from the original parental T24 cell line through continuous exposure to cisplatin of half maximal inhibitory concentration (IC_50_) obtained from the dose–response study of cisplatin exposure for 72 h. Subsequently, the media were removed, and the viable cells were then maintained in the presence of cisplatin. The new IC_50_ values for these cisplatin resistant cells were re-analyzed every 1 month. These cells were then maintained continuously in the presence of cisplatin at the new IC_50_ for an additional 72 h with the repetitive procedure. We finally cultivated the resistant T24 subline (T24/R) at 6 months.

### 4.2. Reagents and Antibodies

Trifluoperazine was obtained from Enzo Biochem (New York, NY, USA) and Cisplatin was obtained from clinical preparations of Abiplatin solution (Pharmachemie BV, Haarlem, the Netherlands). All other chemicals were purchased from Sigma-Aldrich (St. Louis, MO, USA) or Merck Millipore (Billerica, MA, USA). The following antibodies were used for Western blot analysis: Bcl-xL, p21, p27, CHOP, phospho-stress-activated protein kinase (SAPK)/c-Jun N-terminal kinase (JNK, Thr183/Tyr185), and phospho-histone H2A.X (Ser139), which were obtained from Cell Signaling Technology (Danvers, MA, USA). Antibodies against β-actin and glyceraldehyde 3-phosphate dehydrogenase (GAPDH) were purchased from GeneTex (Irvine, CA, USA), and those against JNK and α-tubulin were purchased from Santa Cruz Biotechnology (Santa Cruz, CA, USA). Furthermore, an antibody against caspase-4 was purchased from MBL (Woburn, MA, USA).

### 4.3. Measurement of Cell Viability

The 3-(4,5-dimethylthiazol-2-yl)-2,5-diphenyltetrazolium bromide (MTT) assay (Sigma-Aldrich) was used to detect cell viability. Briefly, the cells were seeded in culture medium in 96-well microplates (5000 cells/well) and incubated at 37 °C for 24 h before drug treatment. The cells were subjected to various treatments for 24 h or 48 h and then incubated in complete medium containing 0.5 mg/mL MTT at 37 °C for 4 h. The reduced MTT crystals were dissolved in dimethyl sulfoxide (DMSO, Sigma-Aldrich), and the absorbance was detected at 570 nm.

### 4.4. Knockdown of BcL-xL Using siRNA

For BcL-xL knockdown, T24/R cells were transfected with 10 nM SMARTpool siRNA targeting BcL-xL (Thermo Scientific Dharmacon, Lafayette, CO, USA) or 10 nM nontargeting scrambled siRNA (as control) by using DharmaFECT 1 transfection reagent (Thermo Scientific Dharmacon) in accordance with the manufacturer’s instructions. Subsequently, the transfected cells were simultaneously cultured with or without chemotherapeutic agents in complete RPMI for 24 h.

### 4.5. Western Blot Analysis

To determine protein expression, the cells were lysed with cell lysis buffer (Cell Signaling Technology) on ice after being with cold phosphate-buffered saline (PBS). The supernatants were collected after centrifugation of cell lysates at 14,000 rpm for 10 min at 4 °C. The bicinchoninic acid protein assay (Thermo Scientific Pierce, Rockford, IL, USA) was used to detect total protein concentrations. Equal amounts of proteins obtained from each group and mixed with sample loading buffer (Biotools, Taipei, Taiwan) were subjected to sodium dodecyl sulfate-polyacrylamide gel electrophoresis and were then transferred onto polyvinylidene fluoride (PVDF) membranes (Merck Millipore). After being blocked with 5% bovine serum albumin (BSA) in PBS, the membranes were incubated with various primary antibodies in PBS at 4°C overnight. After being washed twice with TBST (TBS containing 0.05% Tween 20), the membranes were incubated with horseradish-peroxidase-conjugated secondary antibodies (GeneTex) at recommended dilution ratios in TBST at room temperature for 2 h. The antibody-labeled membranes were again washed twice with TBST and visualized by enhanced chemiluminescence (ECL) substrates (Merck Millipore and Biotools) under an ImageQuant LAS 4000 (GE Healthcare, Chicago, IL, USA) system. The target protein levels, normalized to each internal control, were quantified with Image J software (NIH, Bethesda, MD, USA).

### 4.6. Apoptosis Assay

An apoptosis assay was performed using a Muse Annexin V and Dead Cell Kit (Merck Millipore) in accordance with the manufacturer’s protocol. The stained apoptotic cells were then examined and quantified through flow cytometry (Muse Cell Analyzer, Merck Millipore).

### 4.7. Cell Cycle Analysis by Flow Cytometry

The cells from each cell line were grown in medium as mentioned above. At 40% confluency, the cells were treated with DMSO (as the non-treated control) or trifluoperazine for 24 h. The cells were then collected and processed with a Muse Cell Cycle Assay Kit (Merck Millipore) for cell cycle analysis. Cell cycle distribution was then analyzed using a Muse Cell Analyzer flow cytometry (Merck Millipore).

### 4.8. Combinative Drug Effects

The combinative effect of cisplatin and TFP was determined by using CalcuSyn software (version 1.1.1, 1996, Biosoft, Cambridge, UK). The combinative effect at a combination ratio (TFP:cisplatin = 1:1.25) was subjected as previously described [33,34,35]. The combination index values of less than 1, equal to 1, and greater than 1 were defined as synergism, additive, and antagonism, respectively.

### 4.9. In Vivo Xenograft Experiments

A total of 18 mice were used in this study. T24/R cells (5 × 10^6^) were suspended in 100 μL of serum-free media and mixed with an equivalent volume of Matrigel (BD Biosciences, San Jose, CA, USA). The mixture was subcutaneously injected into the dorsal flanks of 6–8-weeks-old nude mice (obtained from the Taiwan National Laboratory Animal Center, Taipei, Taiwan). The mice were intraperitoneally administered with a mixture of DMSO and normal saline (nontreated control, *n* = 4), TFP (40 mg/kg, three times per week, *n* = 5), cisplatin (10 mg/kg, three times per week, *n* = 5), or a combination of cisplatin with TFP (*n* = 5) after the tumors had grown to approximately 100–150 mm^3^. The tumor sizes were measured using calipers every 4 days, and tumor volume was calculated as follows: longest tumor diameter × (shortest tumor diameter)^2^/2. After 4 weeks of treatment, the tumors were abscised and were photographed. All studies involving animal experiments were approved by the National Taiwan University College of Medicine and College of Public Health Institutional Animal Care and Use Committee (IACUC) (No. 20170557). All animal care and experimental procedures were performed in accordance with protocols approved by the National Taiwan University College of Medicine and College of Public Health IACUC. All studies involving animals complied with the Animal Research: Reporting of In Vivo Experiments (ARRIVE) guidelines for the reporting of experiments involving animals.

### 4.10. Statistical Analysis

Statistical analyses were performed using GraphPad Prism 7 software (GraphPad Software, San Diego, CA, USA), with all data presented as means ± standard deviations or standard errors of the means. Analysis was undertaken using Student’s t-test, with *p* < 0.05 considered statistically significant.

## 5. Conclusions

In summary, TFP enhances the cytotoxicity of cisplatin and alleviated drug resistance in cisplatin-resistant UCs, which may be mediated by downregulation of Bcl-xL. These results provide important insight in clinical applications to find chemosensitizers to augment and improve the therapeutic efficacy of cisplatin-resistant UCs.

## Figures and Tables

**Figure 1 ijms-20-03218-f001:**
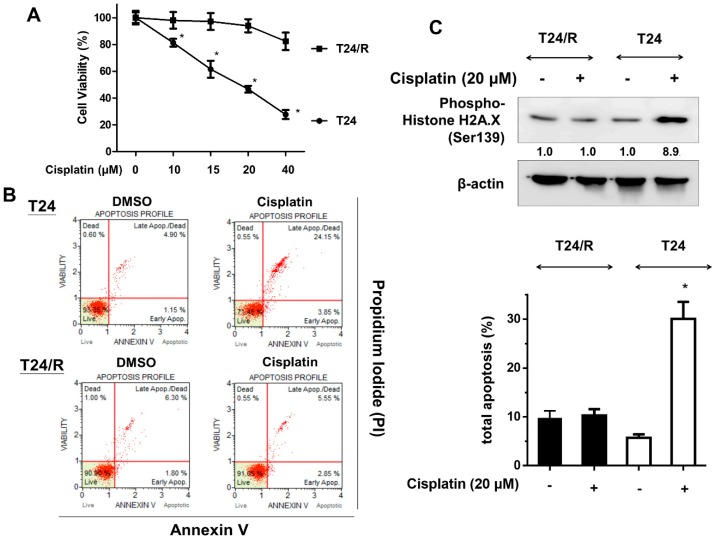
Cisplatin-induced cytotoxicity, apoptosis, and DNA damage were reduced in cisplatin-resistant urothelial carcinoma (UC) (T24/R) cells. (**A**) Parent T24 and cisplatin-resistant UC cell lines (T24/R) were treated with various concentrations of cisplatin (10–40 μM) for 24 h. Cell viability was assessed using the MTT assay. * *p* < 0.05 as compared T24/R cells with T24 cells. (**B**) Cells were exposed to cisplatin (20 μM) and DMSO for 24 h. Apoptotic cells were analyzed through FACS flow cytometry with propidium iodide and annexin V-FITC staining. Data are presented as means ± SD, * *p* < 0.05 as compared with T24/R. (**C**) Cell lysates were harvested, and the expression of a DNA damage marker (phospho-histone H2A.X, Ser139) was assessed using western blot analysis. All results shown are representative of at least three independent experiments.

**Figure 2 ijms-20-03218-f002:**
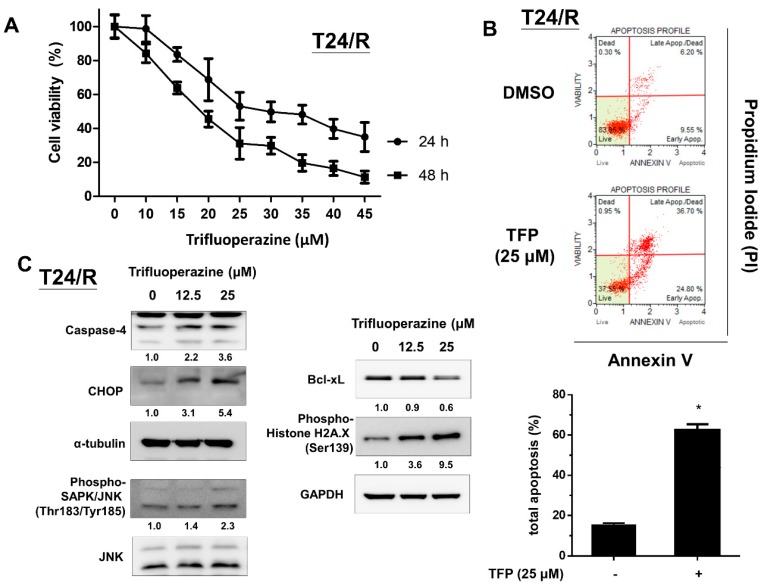
Trifluoperazine (TFP) effectively induced cytotoxicity, apoptosis, endoplasmic reticulum (ER) stress-related apoptosis, and DNA damage in T24/R cells. (**A**) Cisplatin-resistant UC cell lines (T24/R) were treated with mock (DMSO) and various concentrations of TFP (10–45 μM) for 24 h. Cell viability was assessed using MTT assay. (**B**) T24/R cells were separately treated with TFP (25 μM) and DMSO for 24 h. Apoptotic cells were analyzed using FACS flow cytometry with propidium iodide and annexin V-FITC staining. Data are presented as means ± SD, * *p* < 0.05 as compared with mock. (**C**) Cell lysates were harvested and then assessed through Western blot analysis with specific antibodies to cell stress-related molecules phospho-SAPK/JNK (Thr183/Tyr185), ER stress-related apoptosis molecules (CHOP and caspase-4), and a DNA damage marker (phospho-histone H2A.X, Ser139). Results shown are representative of at least three independent experiments.

**Figure 3 ijms-20-03218-f003:**
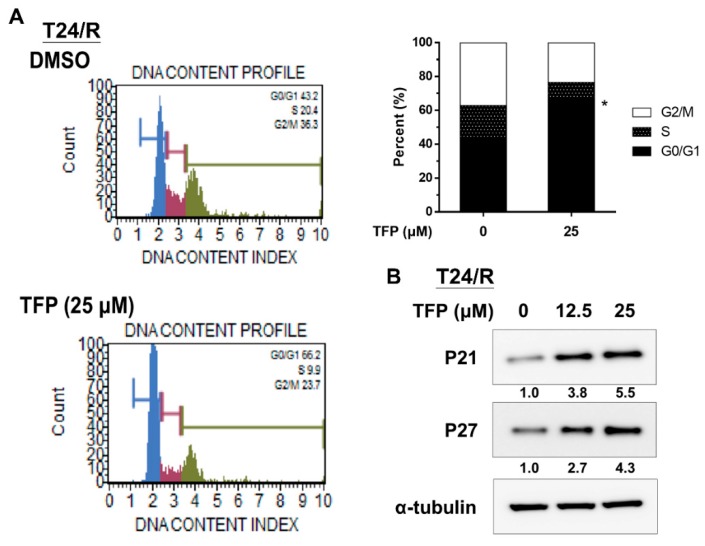
TFP induced G0/G1 arrest in T24/R cells. (**A**) T24/R cells were separately treated with TFP (25 μM) and DMSO for 24 h. Cell cycle analyses were performed through flow cytometry with propidium iodide staining. Quantitative data are presented as means ± SD of three independents experiments, * *p* < 0.05 as compared with control. (**B**) T24/R cells were treated with TFP (12.5 or 25 μM) and DMSO for 24 h. The total cell lysates were assessed for the cyclin-dependent kinase inhibitors (CKIs): p21 and p27 by using Western blot analysis. Results shown are representative of at least three independent experiments.

**Figure 4 ijms-20-03218-f004:**
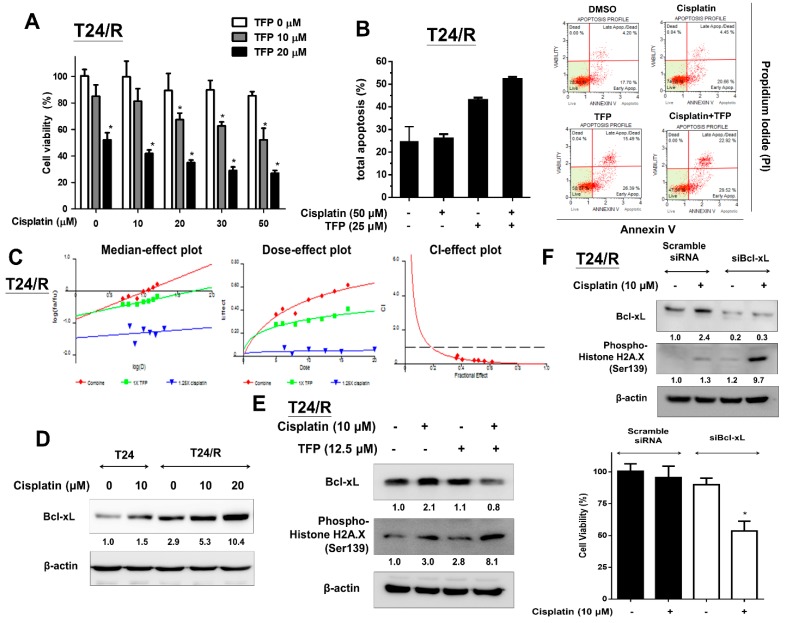
TFP enhanced the antitumor effects of cisplatin on T24/R cells. The alleviation of cisplatin resistance was associated with concurrent suppression of Bcl-xL. (**A**) T24/R cells were treated with cisplatin (10–50 μM) or TFP (10 and 20 μM) alone or in combination for 24 h. Cell viability was determined using the MTT assay. (**B**) Cells were exposed to cisplatin (50 μM) or TFP (25 μM) alone or in combination for 24 h. Apoptotic cells were analyzed through FACS flow cytometry with propidium iodide and Annexin V-FITC staining. (**C**) T24/R cells were incubated in the presence of TFP, cisplatin, and in combination at the concentration ratio of 1:1.25 (TFP:cisplatin). Cell viability was measured by MTT assay after 24 h exposure. The median-effect plot, dose-effect plot and the combination index (CI)-effect plot for TFP, cisplatin, and the combination. The combination of TFP and cisplatin exhibited synergistic effects (combination index <1) in T24/R cells. (**D**) T24 parental cells and T24/R cells were treated with cisplatin (10 μM) and TFP (10–20 μM) separately. Cell lysates were collected and analyzed for Bcl-xL expression using Western blot analysis. (**E**) T24/R cells were treated with cisplatin (10 μM) or TFP (10–25 μM) alone or in combination for 24 h. Cell lysates were subjected to Western blot analysis of Bcl-xL and phospho-histone H2A.X (Ser 139). (**F**) T24/R cells were transfected with scrambled and Bcl-xL siRNA for 24 h, followed by cisplatin (10 μM) treatment for 24 h. Cell viability was determined using the MTT assay. Quantitative analyses of cell viability are presented as the means ± SD. The results shown are representative of at least three independent experiments. * *p* < 0.05 represents a significant difference between the indicated groups.

**Figure 5 ijms-20-03218-f005:**
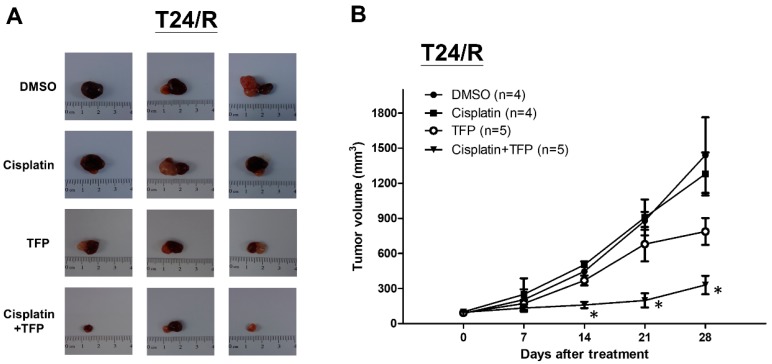
TFP enhanced antitumor effect of cisplatin in T24/R xenograft mouse model. Nude mice bearing cisplatin-resistant T24/R UC xenograft tumors were treated with DMSO (nontreated control, *n* = 4), cisplatin (*n* = 4), TFP (*n* = 5), or a combination of cisplatin and TFP (*n* = 5) for 4 weeks. (**A**) Tumor images representing excised tumors from each group. (**B**) Tumor volume for each group during the 4-week treatment. The data are presented as means ± standard error of the mean. * *p*  <  0.05 represents a significant difference between the cisplatin group and the combination group.

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
