# Peer review of "Trifluoperazine, an Antipsychotic Drug, Effectively Reduces Drug Resistance in Cisplatin-Resistant Urothelial Carcinoma Cells via Suppressing Bcl-xL: An In Vitro and In Vivo Study"

_ijms, 2019, doi:10.3390/ijms20133218_

Round 1

Reviewer 1 Report

This is a well presented study, the results might have latter applicability in clinical settings. I only have minor comments:

The authors should discuss how the experimental setup has relevance for the clinical setting and at what extend. Furthermore, the authos must improve the quality of their figures, especially the FACS results are not readable in this form (e.g. Fig. 1B; 2B; 3A&B; 4B).

Reviewer 2 Report

In the present manuscript Kuo et al. describe cytotoxicity effect, apoptosis and G0/G1 arrest on cisplatin-resistant bladder urothelial carcinoma (UC) after trifluoperazine (TFP) treatment.  Moreover they describe that the antiapoptotic factor, B-cell lymphoma-extra large (Bcl-xL) is upregulated in cisplatin-resistant UC cells (T24/R) and that Bcl-xL knockdown resensitized T24/R to cisplatin cytotoxic effect.  At last, they perform combination studies with TFP and cisplatin in vitro and in vivo.  The manuscript is interesting and well written but needs major improvements.

-         The drug combination study could be improved by the use of software (like Calcusyn, CompuSyn) that calculate the combination index and isobologram of two drugs, indicating if drug combination has a synergic or additive effect.

-         The authors should insert a statistical analysis in figure 5B and indicate if treatments with TFP and cisplatin produce any apparent toxicity in mice.

-         The authors should add WB expression quantification and statistical analysis.

Minor points:

-         The introduction can be developed by more information on TFP antitumor effects and molecular mechanism in cancer.

-         the authors should add references to Fig 4D and 4E in the text

-         the authors could merge the paragraphs 2.4 and 2.5.

Round 2

Reviewer 2 Report

the authors' review improved the manuscript making it suitable for publication